# Preventing pressure injury in nursing homes: developing a care bundle using the Behaviour Change Wheel

Jacqueline F Lavallée,[1] Trish A Gray,[2] Jo C Dumville,[1] Nicky Cullum[3]

[1]Division of Nursing, Midwifery and Social Work, The University of Manchester, Manchester, UK
[2]Division of Nursing, Midwifery and Social Work, School of Health Sciences, Faculty of Biology, Medicine and Health, The University of Manchester, Manchester, UK
[3]School of Nursing, Midwifery and Social Work, The University of Manchester, Manchester, UK

**Correspondence to**
Dr Jacqueline F Lavallée;
jacqueline.lavallee@manchester.ac.uk

## ABSTRACT

**Objective** To develop, with nurse specialists and nursing home care staff, a theory and evidence-informed *pressure injury prevention* care bundle for use in nursing home settings.

**Design** The development of a care bundle.

**Methods** We undertook a detailed, multistaged and theoretically driven development process. First, we identified evidence-informed pressure injury prevention practices: these formed an initial set of possible target behaviours to be considered for inclusion in the bundle. During a 4-hour workshop and supplemental email consultation with a total of 13 healthcare workers, we agreed the key target behaviours for the care bundle. We explored with staff the barriers and facilitators to prevention activity and defined intervention functions and behaviour change practices using the Behaviour Change Wheel.

**Setting** North West England.

**Results** The target behaviours consisted of three elements: support surfaces, skin inspection and repositioning. We identified capability, opportunity and reflective motivation as influencing the pressure injury prevention behaviours of nursing home care staff. The intervention functions (education, training, modelling) and behaviour change techniques (information about social and environmental consequences, information on health consequences, feedback on behaviour, feedback on the outcome of behaviour, prompts/cues, instruction on how to perform the behaviour, demonstration of behaviour) were incorporated into the care bundle.

**Conclusion** This is the first description of a pressure injury prevention care bundle for nursing homes developed using the Behaviour Change Wheel. Key stakeholders identified and prioritised the appropriate target behaviours to aid pressure injury prevention in a nursing home setting.

## BACKGROUND

Pressure injuries are areas of localised damage to the skin and underlying tissue.[1] They are caused by prolonged, or short but intense, periods of pressure or pressure and shear. Pressure injury can lead to severe pain and distress, poor health-related quality of life and serious complications such as gangrene and mortality.[2–4]

> **Strengths and limitations of this study**
>
> ► This study will inform the development of a novel intervention to support nursing home care staff to prevent pressure injury in residents.
> ► Integrating theory, research evidence and expert opinion into the care bundle should maximise the intervention's acceptability, feasibility and potential effectiveness.
> ► The pressure injury prevention care bundle is described in detail along with the intervention's potential mechanisms of action and the specific behaviour change techniques enhancing applicability and reproducibility.
> ► A number of experienced staff participated in the Nominal Group technique, but there was a limited number of tissue viability nurses who participated face-to-face.

Reducing and eliminating pressure injuries across all healthcare settings in the UK are priorities.[5] People at high risk of pressure injury include those who are seriously ill, the elderly and those with impaired mobility.[6 7] Thus, many people living in nursing homes are likely to be at an increased risk of pressure injury. Moreover, a point prevalence survey of complex wounds (eg, pressure ulcers, leg ulcers) conducted in a northern UK city found that 26% of individuals with a pressure ulcer (an open wound caused by pressure) lived in residential or nursing homes.[8]

Pressure injury prevention processes are shaped by national and international guidelines based on a synthesis of research findings and expert opinion.[1 9] Current guidelines recommend a range of clinical interventions including: *risk assessment, skin assessment, repositioning, correction of compromised hydration and nourishment,* the use of *pressure redistributing devices* and *barrier creams, training for care staff* and *accurate monitoring and documentation.* However, the implementation of pressure injury prevention activities remains challenging, particularly

in nursing homes where understaffing, high staff turnover and a lack of monitoring can result in limited staff knowledge and inconsistent clinical care.[10 11]

Care bundles were first introduced by the Institute for Healthcare Improvement to improve the quality and consistency of care.[12] Care bundles comprise three to five evidence-informed clinical interventions (referred to as 'elements'), which have the potential to improve patient outcomes when performed collectively and reliably. The Institute for Healthcare Improvement suggests that every eligible patient should receive all of the bundle elements unless medically contraindicated.[12]

Care bundles aim to change the behaviour of healthcare workers; therefore, the use of behaviour change theory is key.[13] While several care bundles have been developed, it is not always clear how they were developed or whether they were underpinned by theory.[14] There are multiple theories and frameworks for behaviour change, many with overlapping constructs.[15 16] The Behaviour Change Wheel[15 17] is a framework for designing behaviour change interventions and was developed to facilitate the integration of target behaviours, behaviour change theory and intervention development through a series of three key stages that can be subdivided into eight steps (online supplementary appendix 1). Thus, the Behaviour Change Wheel outlines a systematic and transparent approach to identify the appropriate theory-based intervention content that may bring about change in the people who are its target (in this case, nursing home staff).

The COM-B model[17] forms the centre of the Behaviour Change Wheel[15 17] and assists with understanding the behaviour in context (stage 1 of intervention development). The COM-B model hypothesises that capability (C), opportunity (O) and motivation (M) all interact and can explain behaviour (B) and can become the focus for the behaviour change intervention. Within the COM-B model, *capability* refers to the person's psychological and physical capacity to engage in the target behaviour. *Opportunity* refers to the factors that are external to the individual and influence the potential success of the behaviour (ie, the physical environment or the social environment). *Motivation* involves the psychological processes that can trigger and direct behaviour, including reflective and automatic motivation.

Once the targets for change (eg, physical opportunity) have been identified using the COM-B model, the second and third stages of the Behaviour Change Wheel focus on how intervention developers might facilitate change in these areas using intervention functions, policy categories, behaviour change techniques and modes of delivery. It is recommended that developers consider their intervention design using the APEASE criteria.[15] The APEASE criteria are used to guide the decisions on the intervention content and how to implement the intervention within a particular setting.[15 17] These criteria involve an assessment of affordability, practicability, effectiveness and cost-effectiveness, acceptability, side-effects/safety and equity.

We were unable to identify any pre-existing pressure injury prevention care bundles designed for, and implemented in, nursing home settings. All of the published pressure injury prevention care bundles focus on acute hospital settings such as intensive care units and critical care units.[18–23] This paper describes the development of the first reported nursing home-specific pressure injury prevention care bundle. We aimed, with key stakeholders from nursing homes and the National Health Service (NHS), to coproduce a pressure injury prevention care bundle that is relevant to the nursing home context. We describe how the Behaviour Change Wheel was used to support the theory-driven processes in the design of the implementation plan for the care bundle. Figure 1 presents a logic model illustrating our knowledge and understanding at the start of this work and the outcomes we were aiming for. At the end of the work, we aimed to design the components of the intervention (the 'solution' in figure 1).

## METHODS
### Study design
We describe a two-part care bundle development process. Part 1 used the Nominal Group technique[24] to gain consensus about the elements to include in the care bundle. Part 2 followed the steps outlined in the Behaviour Change Wheel to facilitate the development of the implementation plan for the care bundle.

### Participants
The study took place in the North West of England. Purposive sampling was used to recruit participants with relevant clinical and management experience and expertise. Participants were eligible to participate if they were a nursing home-based registered nurse (referred to from now on as a nurse), manager or healthcare assistant or a community-based tissue viability nurse. Written consent was gained from all participants.

### Materials and procedures
Figure 2 presents a diagrammatical outline of the processes involved in developing the care bundle and how we applied the Behaviour Change Wheel processes here.

#### Stage1: understanding the behaviours

*Behaviour Change Wheel step 1: defining the problem in behavioural terms (preworkshop)*
We reviewed the pressure injury prevention literature to gain an understanding of the main barriers to pressure injury prevention in nursing homes. We conducted a systematic review that identified and explored existing care bundles and any evidence for particular design features and behaviour change approaches that might be associated with positive clinical outcomes.[14]

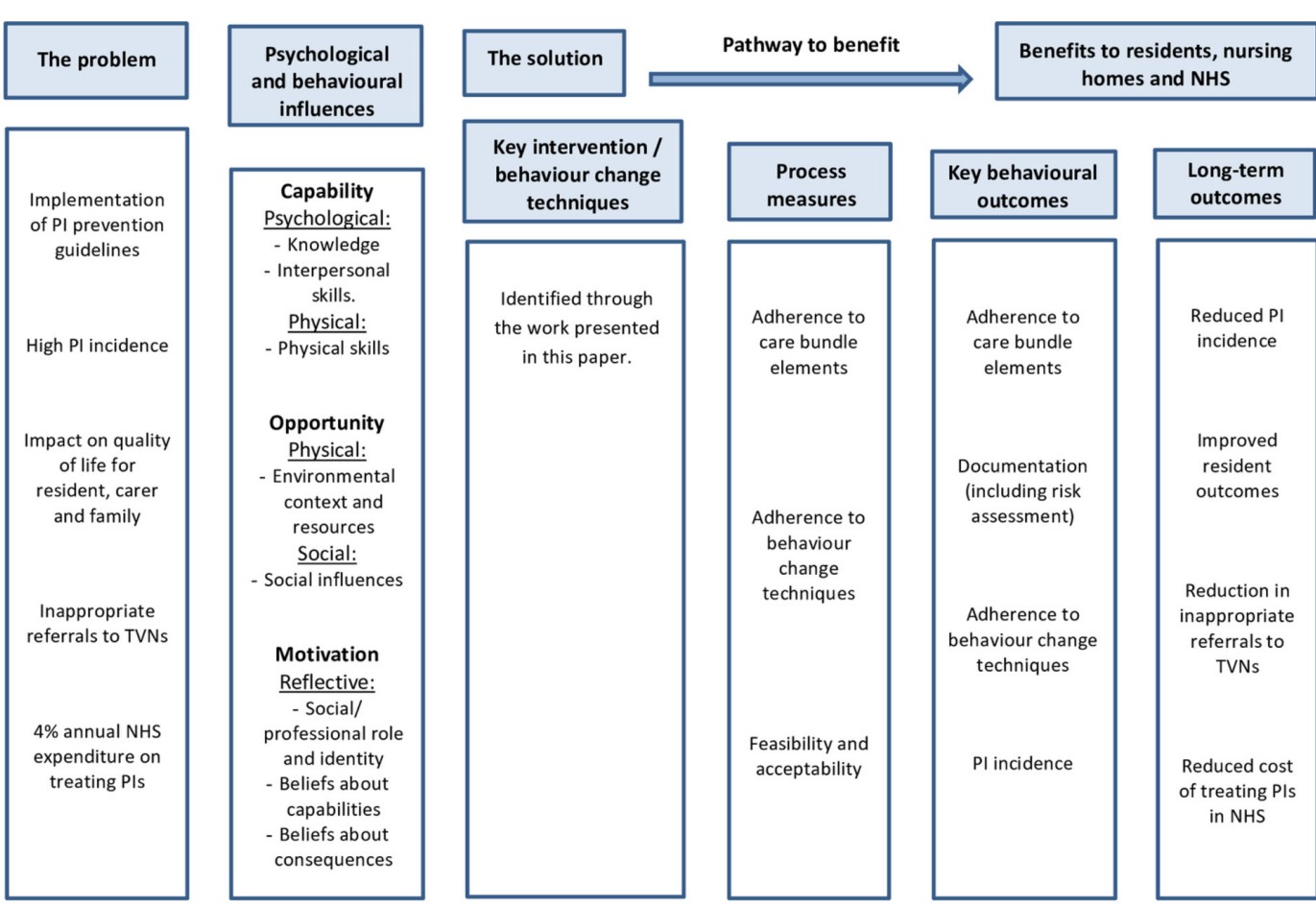

**Figure 1** Logic model for the pressure injury prevention care bundle outlining the consequences of pressure injury in nursing homes, the potential behavioural causes of pressure injury and the pathway to benefit through preventing pressure injury. NHS, National Health Service; PI, Pressure Injury; TVN, tissue viability nurse.

*Behaviour Change Wheel steps 2 and 3: selecting and specifying the target behaviours (care bundle development workshop)*

These two steps involved the identification of care bundle elements (ie, the specific pressure injury prevention clinical interventions) and consideration of who, what, when, where and how often the care bundle elements should be delivered. We held a 4-hour interactive workshop with key stakeholders to identify the clinical interventions to assist with pressure injury prevention in nursing homes. There are several possible methods that can be drawn on for developing a care bundle. The Nominal Group technique was developed to facilitate the decision making of groups.[24] In essence, we used the Nominal Group technique to gain consensus about the most important pressure injury prevention elements to be included in the care bundle. This approach is highly structured, usually delivered face-to-face, consisting of multiple rounds where items or questions are rated, discussed and rerated by the expert panellists (eg, nurses). This method minimises the effects of any dominant participants as all group members are provided with equal opportunities for voting.

We presented participants with an overview of the research-based international and national pressure injury prevention guidelines.[1 9] We then discussed the guideline recommendations, focusing in particular on their applicability in a nursing home setting. All participants had the opportunity to add any clinical interventions they thought were missing from the guidelines before they began voting.

The Nominal Group process was explained and participants were split into two groups for voting purposes (ie, healthcare assistants or registered nurses). Each participant within these groups was given five votes in the form of coloured stickers, which they used to vote individually for their top three to five pressure injury prevention clinical interventions. The colour of the sticker indicated whether the voter was a nurse or healthcare assistant. We counted the votes in real time and presented the results to the participants to facilitate discussion prior to the second round of voting. In the case of a tie, we offered the participants one extra vote for one of the two tied clinical interventions. We invited participants to express their opinions on the clinical interventions and whether they believed clarification was required. Again, colour-coded stickers were used to cast votes in the second round. This round was used to finalise the agreement between participants.[24] The care bundle elements were agreed after a final discussion of the clinical interventions that received the highest numbers of votes.

**Behaviour Change Wheel stage 1: understand the behaviour**

**Figure 2** Data collection and analysis processes used to develop the care bundle using the steps and stages outlined in the Behaviour Change Wheel (BCW). *Methods and findings reported in the study by Lavallée et al.[14] **Methods and findings reported in the study by Lavallée et al.[25]

We then asked the workshop participants to specify the detail for each bundle element; the frequency with which they should be delivered, where and by whom and we asked participants to score the components of each element out of 10 (0=not important, 10=extremely important). Following the workshop, the care bundle elements and specific components were reviewed in line with existing research evidence and cross-checked for validity by experts in the field such as tissue viability nurses.

*Behaviour Change Wheel step 4: identifying what needs to change to enable the reliable delivery of pressure injury prevention clinical interventions*

We purposively recruited individuals who provide care for those at risk of developing pressure injuries in nursing homes and collected data from 25 participants (healthcare assistants (n=7), registered nurses (n=11), nurse managers (n=3) and community-based tissue viability nurses (n=4)). Using semistructured interviews, we explored the barriers and facilitators to pressure injury prevention[25] using the Theoretical Domains Framework.[26] The Theoretical Domains Framework comprises 14 domains that can be used to explore the determinants of professional behaviour change and inform intervention design (eg, knowledge, social influences, beliefs about consequences).[26] Each of the 14 Theoretical Domains Framework domains can be mapped onto the COM-B model[15 17] to facilitate the understanding of healthcare workers' behaviours within a particular context.

We analysed the data deductively, using the Theoretical Domains Framework and identified the behavioural and psychological influences on pressure injury prevention by mapping the salient barriers and facilitators identified onto the COM-B model, using the guidance provided by the Behaviour Change Wheel.[15]

### Stages 2 and 3: identifying the intervention content and implementation options
*Behaviour Change Wheel steps 5–8: identifying the intervention functions, policy categories, behaviour change techniques and modes of delivery*

We mapped those components of the COM-B model identified as being relevant to pressure injury prevention in nursing homes (step 4) onto the matrices provided in the Behaviour Change Wheel, and this informed our plan for implementing the care bundle. In addition, the Behaviour Change Technique Taxonomy V.1[27] informed our choice of behaviour change techniques (step 7). The Behaviour Change Technique Taxonomy V.1[27] comprises 93 behaviour change techniques and can be used to identify intervention components, enabling the standardisation of terms as well as the comparison of behaviour change techniques across studies. We applied the APEASE criteria[15] for designing and evaluating interventions to each of the relevant implementation aspects to guide our judgements in selecting the most appropriate intervention functions, policy categories, behaviour change

techniques and modes of delivery likely to support the successful implementation of the care bundle.

To ensure the implementation plan was suitable, we held discussions individually with the nursing home care staff, tissue viability nurses and academic researchers before we finalised the care bundle. These discussions were based on the 'modelling' guidance provided by the UK Medical Research Council's guidance for developing and evaluating complex interventions,[13] which includes who should receive the intervention, how changes to practice are usually introduced, what the barriers to change might be and how delivery can be documented.

### Patient and public involvement

Nursing home residents and the public were not involved in the development of the care bundle.

## RESULTS
### Behaviour Change Wheel stage 1: understanding the behaviours
#### Behaviour Change Wheel step 1: defining the problem in behavioural terms (preworkshop)

Our review of the literature identified that understaffing, high turnover and limited staff knowledge are commonly reported as barriers to pressure injury prevention[10 11] and that good communication and positive attitudes to pressure injury prevention are described as facilitators.[28–30] In addition, central to the prevention of pressure injuries is the belief that the actions of healthcare workers (eg, repositioning) directly influence the development of pressure injuries.[31] Consequently, care bundles may be an effective tool to improve the implementation of guidelines and evidence-informed practices.[14]

Within our systematic review, we were not able to conduct a meta-regression of study features or explore the magnitude of effects as there were insufficient comparisons involving patient outcomes. Consequently, we conducted subgroup analyses. We found that all care bundles (regardless of the number of elements) reduced the risk of the negative patient outcomes and the apparent effect of care bundles appeared to reduce as the number of elements increased. The lowest risk for the negative patient outcomes was in the subgroup with 'eight behaviour change techniques'. However, we considered these data to be of very low quality. Our findings from the systematic review are reported in detail elsewhere.[14]

#### Behaviour Change Wheel steps 2 and 3: selecting and specifying the target behaviours (care bundle development workshop)

Ten participants attended the workshop, including one tissue viability nurse and staff from one nursing home (four healthcare assistants and five registered nurses). A further three tissue viability nurses were unable to attend the workshop but participated in email (n=2) or face-to-face (n=1) consultations, which followed the processes outlined in the Methods section as closely as possible. The participants' ages ranged from 26 to 55 years, one participant was male and one had previously attended wound care training. The median years of experience in working with people at risk of developing pressure injuries was 11 years (IQR: 1.4–13 years).

During the discussion prior to round 1, it was agreed that '*pain management*' should be added as a clinical intervention and *nutrition and hydration* should be separated into two. The clinical interventions voted for in round 1 by each group differed (table 1). For example, the healthcare assistants did not vote for *skin assessment*, whereas 80% of the nurses (4/5) and 75% of the tissue viability nurses (3/4) did. Similarly, 75% of the healthcare assistants (3/4) and 50% of the tissue viability nurses (2/4) voted for *support surfaces* to be included but the nurses did not. During the discussion, the nurses explained that they did not select *support surfaces* as a key clinical intervention as they felt that pressure redistributing devices covered this (although this only received one vote from the nurses' group). Further discussion resulted in reuniting *nutrition and hydration* as all nursing home participants explained that they offer these together. Consequently, six clinical interventions went through to the second round of voting (*skin care, continence care, skin assessment, repositioning, nutrition and hydration* and *support surfaces*).

*Repositioning, skin assessment, skin care, continence care* and *nutrition and hydration* were voted into the top five in round 2 (table 1). Every tissue viability nurse voted for *support surfaces*, whereas the healthcare assistants considered support surfaces to be important but embedded within *repositioning*, and this was reflected in their voting. Through discussion, the participants agreed that including *support surfaces* as an element separate from *repositioning* was important and *support surfaces* should also incorporate *pressure redistributing devices*. While the participants deemed *nutrition and hydration* and *continence care* important, they agreed that only those residents with inadequate nutrition and hydration require additional nutrition and fluid[9]; therefore, this element would be redundant for some individuals (making the care bundle more of a checklist). Participants believed that continence care was a separate, complex issue, requiring a number of detailed steps to prevent damage to skin integrity and likely to require its own care bundle.[32] Consequently, participants decided that providing and monitoring such clinical interventions are part of basic care and should not be included in a specific pressure injury prevention bundle. The *skin care* and *skin assessment* clinical interventions were merged and three elements made up the care bundle: *support surfaces, skin inspection* and *repositioning*.

Participants ranked, in order of perceived importance, the components required to ensure the accurate and consistent completion of each of the care bundle elements. All participants agreed that residents should receive a monthly pressure injury risk assessment to trigger the activation of the care bundle for those at risk of developing a pressure injury. However, more frequent assessments may be warranted for some residents at high risk of pressure injury development or if there is a change

**Table 1** Votes from rounds 1 and 2 from each healthcare staff group

| Clinical intervention | Healthcare assistants (n=4) | Nurses (n=5) | Tissue viability nurses (n=4) | Overall percentage of votes |
|---|---|---|---|---|
| Voting round 1 | | | | |
| Nutrition | 1 | 4 | 4 | 69 |
| Hydration | 2 | 4 | 0 | 31 |
| Skin care | 2 | 1 | 1 | 38 |
| Support surfaces | 3 | 0 | 2 | 46 |
| Repositioning | 3 | 5 | 4 | 92 |
| Continence care | 4 | 5 | 4 | 100 |
| Pressure redistributing devices | 1 | 1 | 2 | 31 |
| Skin assessment | 0 | 4 | 3 | 54 |
| Pain | 0 | 0 | 0 | 0 |
| Barrier cream | 0 | 0 | 0 | 0 |
| Voting round 2 | | | | |
| Skin care | 2 | 4 | 1 | 45 |
| Continence care | 4 | 4 | 4 | 92 |
| Skin assessment | 0 | 3 | 3 | 46 |
| Repositioning | 4 | 5 | 4 | 100 |
| Nutrition and hydration | 4 | 5 | 4 | 100 |
| Support surfaces | 0 | 0 | 4 | 31 |

in a resident's clinical status. The frequency with which the elements of care are to be delivered will be informed by the risk assessment, although the risk assessment was separate from the care bundle. It was agreed that the nursing home care staff should complete and document every element of the care bundle for all residents deemed to be at risk of developing a pressure injury, and where an element cannot be completed a reason must be provided (eg, where a resident has refused to be repositioned).

### Behaviour Change Wheel step 4: identifying what needs to change to enable the reliable delivery of pressure injury prevention clinical interventions

The semistructured interview data (reported elsewhere[25]), when mapped on to the COM-B model, suggested the following factors as influences on the prevention of pressure injury in nursing home settings: *psychological and physical capability; physical and social opportunity*; and *reflective motivation*. We found that improvements in pressure injury prevention knowledge and skills are required. In particular, the tissue viability nurses could provide information about, and training on, pressure injuries and how to prevent them within a nursing home context; but the nursing home care staff need to be permitted to attend this training. In addition, there appears to be scope to increase the use and documentation of evidence-informed pressure injury prevention interventions. Pressure injury prevention interventions need to be conducted in line with the resident's risk of developing a pressure injury. If it is not possible to complete an aspect of care, this must be documented.

### Behaviour Change Wheel stage 2: identifying the intervention content and implementation options

We used the Behaviour Change Wheel to define the key intervention functions and policy categories that could be used to improve pressure injury prevention in nursing homes using the relevant COM-B components identified in step 4.

### Step 5: intervention functions

The three most suitable intervention functions were *education, training* and *modelling* (ie, providing a role model such as a skin champion). Increasing the knowledge of the nursing home care staff and improving their skills through education and training is a crucial aspect to facilitating the prevention of pressure injury in nursing home residents. The inclusion of skin champions should assist with accessing training and education as these can be delivered in-house by the skin champion.

### Step 6: policy categories

The policy categories most suitable for achieving the behaviour change included *communication/marketing* (eg, posters), *guidelines, regulation* and *service provision*.

### Step 7: behaviour change techniques

Using the Behaviour Change Technique Taxonomy V.1[27] (which is a taxonomy of 93 behaviour change techniques) together with the findings from our systematic review, we selected the seven techniques we believed were most suitable to facilitate behaviour change and support prevention practices (*information about social and environmental*

*consequences; information on health consequences; feedback on behaviour; feedback on the outcome of the behaviour; prompts/ cues; instruction on how to perform the behaviour; demonstration of behaviour).*

### Step 8: mode of delivery

We then formulated a plan regarding how and by whom the care bundle would be implemented in practice, and this was based on the discussions held with key stakeholders. The delivery of the care bundle will differ at specific stages, and the key modes of delivery are specified in table 2 (eg, the tissue viability nurses will deliver the face-to-face group training to address the *capability* of nursing home care staff as identified through the COM-B model in stage 1).

## DISCUSSION

This is the first explicit, behaviour change theory-driven, pressure injury prevention care bundle that we have been able to identify. We identified the important elements of the care bundle in collaboration with key stakeholders. Using the COM-B model and with the steps outlined in the Behaviour Change Wheel, we developed a pressure injury prevention care bundle that focused on the three identified target behaviours (*checking of support surfaces, skin inspection* and *repositioning*). The broad functions of the intervention (*education, training, modelling*) aim to be achieved using seven theoretically based behaviour change techniques delivered using a variety of methods, including face-to-face and written materials. This information can be used to populate the solutions box in figure 1 in the introduction (figure 3).

Three main aspects of pressure injury prevention that consistently feature in care bundles were included within our nursing home care bundle, although operationalised differently: repositioning, skin assessment and the use of support surfaces.[18–20 33] However, our care home-focused intervention differs from those delivered in hospital settings as we did not incorporate continence care or nutrition and hydration; mainly because they were deemed core aspects of nursing care that should be prioritised irrespective of any tenuous link with pressure injury prevention. While our care bundle elements reflect those included in hospital-focused bundles, the process of deciding how to promote the behaviour changes around these target behaviours has not been clear in previous work. We supported this work using a strong theoretical framework for intervention design. Through the transparent reporting of the mechanisms of action, modes of delivery and the theoretical constructs, future evaluations of the effectiveness of this care bundle will be possible.

### Strengths and limitations

The theoretical basis and systematic presentation of the development of the care bundle are strengths of our study. The empirical work revealed the target behaviours required (ie, checking of support surfaces, skin inspection, repositioning), and the Behaviour Change Wheel identified the implementation interventions suitable for the care bundle. Previous studies detailing pressure injury prevention care bundles[18 20] have not provided such explicit and transparent methods, which may limit the understanding of the mechanisms of action and causal relationships within the interventions.[34] Thus, the present study addresses these concerns, facilitating subsequent evaluations and future replications.

The use of the Nominal Group technique to develop the care bundle was beneficial for many reasons. The participation of the nursing home care staff and the NHS tissue viability nurses was vital to ensure the integration of specialist knowledge alongside context-specific expertise. The Nominal Group technique enabled each participant to express their view (via individual votes), which minimised the effects of any potentially dominant participants. Using the Nominal Group technique during the workshop was advantageous as it yielded extensive and rich data in a relatively short period of time.

A limitation was the exclusion of residents and their families, as well as the wider multidisciplinary team (eg, podiatrists, dieticians) and the inclusion of only one nursing home and the relatively small number of tissue viability nurse workshop participants. Expert opinion is a fundamental aspect of the Nominal Group technique, and while the majority of the participants who did attend had a range of expertise in caring for individuals residing in nursing homes, specialist nurse input was crucial. Initially, all of the local tissue viability nurses agreed to attend; however, due to unforeseen circumstances, some could not. Consequently, the process was repeated with the tissue viability nurses via face-to-face meetings or online consultations to ensure their specialist knowledge of the prevention of pressure injuries could be combined with the results. We believe that taking such a systematic and structured approach to designing the care bundle will result in a more effective intervention and will aid subsequent evaluations and improvements.

### Future research

The next phase of this research is to test the feasibility of implementing the care bundle in a nursing home context. If the care bundle intervention is feasible and acceptable to nursing home care staff, further evaluation will be necessary to assess the clinical effectiveness and cost-effectiveness. The explicit theoretical links provided through the use of the Behaviour Change Wheel[15 17] and Behaviour Change Technique Taxonomy V.1[27] will facilitate future replications and data synthesis. In addition, exploring the views of residents, their families and the wider multidisciplinary team will be vital to ensure that a holistic approach is taken for the prevention of pressure injuries in nursing home residents.

### CONCLUSION

Care bundles have received much attention within inpatient settings over the past decade due to the potentially synergistic effect of incorporating several clinical

**Table 2** Implementation plan for pressure injury prevention care bundle

| What | Why | Who | How/frequency | Where |
|---|---|---|---|---|
| *Training and education:*<br>- On risk factors, pressure injury prevention, equipment, outcomes, protocols. | Access to training was identified as a barrier to pressure injury prevention in nursing homes.<br>To improve pressure injury prevention knowledge and skills in nursing home care staff (registered and unregistered). We identified the following two BCTs as important components of the intervention: information about social and environmental consequences' and 'information on health consequences'. | Provided by a tissue viability nurse to nursing home care staff (registered and unregistered). | Training will be provided 1 week prior to the implementation of the care bundle and will be a one-off face-to-face, 3-hour interactive group session. Presentation using PowerPoint and printed materials will be provided to the staff who attend and also to the nursing home for staff who are unable to attend. Additional training sessions will be offered to the nursing home care staff to maximise attendance. | Due to practical reasons, training will be held off-site. Written training materials will be available in the nursing home. |
| - On the care bundle and each individual element (support surfaces, skin inspection, repositioning) and how to use the care bundle in practice. | To increase the uptake of the care bundle and to familiarise staff with the processes involved. | Provided to nursing home care staff (registered and unregistered) by a researcher with expertise in behaviour change. | Face-to-face 1 hour interactive group session. PowerPoint and printed materials will be provided to staff who attend and also to the nursing home for staff who are unable to attend. | |
| *Modelling and demonstration of behaviour:*<br>- Skin champions | The skin champions will deliver the care bundle as intended and will be available during a shift. Staff can speak with the skin champions if they have any concerns or queries. Skin champions are also able to demonstrate pressure injury prevention techniques and provide examples of good record keeping (ie, documentation). | Nursing home care staff (likely to be a registered nurse). | This is available face-to-face and is likely to be delivered on an individual basis and will be available as required.<br>The researcher will meet with the skin champions at least bi-weekly to discuss any issues or concerns. | Nursing home. |
| *Implementation of the care bundle:* | | | | |
| - Risk assessment | To identify any risk factors for the development of a pressure injury and indicate the frequency with which the care bundle needs to be delivered. | Registered nurse and/or nursing home manager. | Using a validated risk assessment tool, the risk assessment will be completed at least monthly. If there is a change to a resident's clinical status, the risk assessment should be conducted again. | Nursing home. |
| *Implementation of the care bundle:* | | | | |
| - Complete care bundle for each eligible resident (support surfaces, skin inspection, repositioning). | To improve the reliability of care and to prevent pressure injuries using elements identified locally as being important within a nursing home context. To improve the documentation of pressure injury prevention practices. | Nursing home care staff (registered and unregistered). | Nursing home care staff will complete each element of care included within the care bundle. If it is not possible to conduct all of the elements (*support surfaces, skin inspection, repositioning*) within the care bundle, this must be documented on the overleaf section of the care bundle documentation sheet. The frequency with which this needs to be conducted will depend on each individual resident. The frequency should be amended in line with a resident's needs and risk. For example, for those at risk of developing a pressure injury, it should be at least every 6 hours, at least every 4 hours for those at a high risk and at least every 2 hours for those at a very high risk.<br>Staff are required to ensure the appropriate pressure relieving equipment is being used and is functioning. | Nursing home. |

 Lavallée JF, *et al. BMJ Open* 2019;9:e026639. doi:10.1136/bmjopen-2018-026639

**Table 2**   Continued

| What | Why | Who | How/frequency | Where |
|---|---|---|---|---|
| *Prompts and cues* | An aide memoire was reported as a facilitator of pressure injury prevention in nursing homes. Thus, posters will be placed in staff communal areas (eg, nursing office) to remind staff of the steps involved within the care bundle. The care bundle itself also acts as a checklist as staff are required to document the provision of care on the care bundle sheets. | The research team will provide posters and care bundle documentation. | The unit manager will decide the positioning of the posters on the unit (see online supplementary appendix 2). The nursing home staff are responsible for the completion of the care bundle and associated documents. These will be available daily throughout the study period. | Nursing home (including nursing office, residents' bedrooms, residents' files). |
| *Feedback:* | To maintain motivation and engagement with the care bundle. | | | |
| - On behaviours and outcomes. | To highlight areas of pressure injury prevention where staff are maintaining high levels of care and the areas that could be improved. | Researcher | The research team will provide verbal feedback to the unit manager on a monthly basis during the study period. This will include the number of pressure injuries acquired and adherence to the care bundle. Feedback will be provided in the form of percentages on the following: ► All-or-none compliance (when all aspects of the care bundle were delivered, including times when it was not possible to complete the care bundle but reasons were documented); ► Overall adherence with each individual element: support surfaces, skin inspection, repositioning. Following the completion of the study, the above information will be collated and the findings from the whole study period will be presented verbally to the unit manager and nursing home care staff. | Nursing home. |

**Care bundle elements:**

- Support surfaces

- Skin inspection

- Repositioning

**Behaviour change techniques:**

- Information about social and environmental consequences

- Information on health consequences

- Feedback on behaviour

- Feedback on the outcome of the behaviour

- Prompts/cues

- Instruction on how to perform the behaviour

- Demonstration of behaviour

**Figure 3**   Solutions box for figure 1 detailing the content of the pressure injury prevention care bundle and the steps required to implement the care bundle in nursing homes.

interventions within one package. The structure of care bundles can be used to facilitate reliable and sustainable changes in the work habits of staff. However, few theory-informed care bundles are reported within the literature. This paper describes how a pressure injury prevention care bundle was developed for use in UK nursing homes and how the Behaviour Change Wheel guided the development of the intervention. Key stakeholders contributed to the design of the care bundle, forging the first step towards standardising pressure injury prevention practices within nursing home settings. While preventing pressure injuries in nursing home residents is complex and multifaceted, this structured and transparent approach has facilitated a thorough process for the development of the intervention. The next step is to assess the feasibility of implementing this care bundle within the nursing home environment to ensure that it is acceptable before wider evaluation ensues.

**Acknowledgements**   We are grateful to the NIHR CLAHRC Greater Manchester for supporting this work. The NIHR CLAHRC Greater Manchester is a partnership

between providers and commissioners from the NHS, industry and the third sector, as well as clinical and research staff from The University of Manchester. The views expressed in this article are those of the authors and not necessarily those of the NHS, NIHR or the Department of Health and Social Care.

**Contributors** NC had the original research idea. JFL, TAG, JCD and NC conceived the idea and design for the overall project. JFL developed the standard operating procedures for the workshop and held the email/face-to-face consultations with the tissue viability nurses. TAG facilitated the workshop. All authors contributed to the interpretation of study findings, critical revision of the manuscript for important intellectual content and approval of the final manuscript.

**Funding** This project was funded by the National Institute for Health Research Collaboration for Leadership in Applied Health Research and Care (NIHR CLAHRC) Greater Manchester.

**Competing interests** None declared.

**Patient consent for publication** Not required.

**Provenance and peer review** Not commissioned; externally peer reviewed.

**Data sharing statement** All data relevant to the study are included in the article or uploaded as supplementary information.

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
