## [Reviewer comments · BMJ Open]

ARTICLE DETAILS

TITLE (PROVISIONAL)	Preventing pressure injury in nursing homes: developing a care bundle using the Behaviour Change Wheel
AUTHORS	Lavallée, Jacqueline; Gray, Trish; Dumville, Jo C.; Cullum, Nicky

VERSION 1 - REVIEW

REVIEWER	Zena Moore Royal College of Surgeons in Ireland
REVIEW RETURNED	01-Oct-2018

GENERAL COMMENTS	Many thanks for this paper Preventing pressure ulcers in nursing homes: developing a care bundle using the Behaviour Change Wheel. Reading the paper you mention a number of models/techniques: 1. Nominal group technique2. Theoretical domains framework3. Behaviour change wheel4. APEASE5. Com B model6. Behaviour change technique taxonomy version 1. For me, none of these models/techniques are adequately described, and given their importance to the understanding of the exact process employed, I feel that this lack of description is an important limitation of the paper. In order to enhance readability and understanding a succinct description of each should be provided.
--

REVIEWER	Michael Clark Welsh Wound Innovation Centre UK
REVIEW RETURNED	26-Oct-2018

GENERAL COMMENTS	This manuscript reports the development of a care bundle intended to help the prevention of pressure ulcers in UK Nursing Homes. The work is well described with the limitation of the small sample of participants clearly noted. There may be value in extending the background section to provide information upon the occurrence of pressure ulcers in UK nursing homes (where this data exists?).
---

REVIEWER	Donna E. Martin University of Manitoba Canada
REVIEW RETURNED	31-Oct-2018

GENERAL COMMENTS	Thank you for submitting this manuscript about the development of a care bundle to prevent pressure ulcers in residents of a nursing home in North West England. I have several suggestions to strengthen this paper. Recently, the correct term is pressure injuries. Although the title refers to nursing home settings, it is clear that this care bundle development occurred in one nursing home. The study design was the development of a care bundle and this should be consistently presented and revised in the abstract. Additional limitations are exclusion of residents (please see recent Australian studies), residents' families, and multidisciplinary team members such as allied health professionals and physicians. It would be helpful to address the tension between "treating the whole patient rather than the hole in the patient" as care bundles may be interpreted as "treating the hole in the patient" by some. In the section about the Nominal Group process, further details about deciding on three bundle elements (rather than five) and then excluding the most popular elements would be helpful. These choices were perplexing given the percentages in the Table. Please explain why you decided against using an established skin assessment tool. In Step 4, it is unclear who the 25 participants were and how these participants were recruited. Further details about the analysis process with direct quotes would strengthen this section. Please attend to minor grammatical issues with correct use of commas, semi-colons and colons. I look forward to reviewing the revised manuscript and thank you for your commitment to quality care of residents in nursing homes!
--

VERSION 1 – AUTHOR RESPONSE

Reviewer 1	Many thanks for this paper Preventing pressure ulcers in nursing homes: developing a care bundle using the Behaviour Change Wheel. Reading the paper you mention a number of models/techniques: Nominal group technique, Theoretical domains framework, Behaviour change wheel, APEASE, Com B model, Behaviour change technique taxonomy version 1. For me, none of these models/techniques are adequately described, and given their importance to the understanding of the exact process employed, I feel that this lack of description is an important limitation of the paper. In order to enhance readability and understanding a succinct description of each should be provided.	1. Nominal group technique: We have provided some additional information on page 10.	There are several possible methods that can be drawn on for developing a care bundle. The Nominal Group technique was developed to facilitate the decision making of groups [24]. In essence we used the Nominal Group technique to gain consensus about the most important pressure injury prevention elements to be included in the care bundle. This approach is highly structured, usually delivered face-to-face; consisting of multiple rounds where items or questions are rated, discussed and re-rated by the expert panellists (e.g., nurses).	10/ 160- 167
		2. Theoretical domains framework: We have provided more detail on page 12.	Using semi-structured interviews we explored the barriers and facilitators to pressure injury prevention [25] using the Theoretical Domains Framework [26]. The Theoretical Domains Framework comprises 14 domains that can be used to explore the determinants of professional behaviour change and inform intervention design (e.g., knowledge, social influences, beliefs about	12/ 205- 217

			consequences) [26]. Each of the 14 Theoretical Domains Framework domains can be mapped onto the COM-B model [15, 17] to facilitate understanding of healthcare workers' behaviours within a particular context. We analysed the data deductively, using the Theoretical Domains Framework and identified the behavioural and psychological influences on pressure injury prevention by mapping the salient barriers and facilitators identified onto the COM-B model, using the guidance provided by the Behaviour Change Wheel [15].	
		3. Behaviour change wheel: We have added information to this section on pages 6 and 7.	The Behaviour Change Wheel [15, 17] is a framework for designing behaviour change interventions and was developed to facilitate the integration of target behaviours, behaviour change theory and intervention development through a series of three key stages that can be subdivided into eight steps (Appendix 1). Thus, the Behaviour Change Wheel outlines a systematic and transparent approach to identify	6-7/ 81-88

			the appropriate theory-based intervention content which may bring about change in the people who are its target (in this case, nursing home staff).	
		4. APEASE: We have provided additional information on page 7.	It is recommended that developers consider their intervention design using the APEASE criteria [15]. The APEASE criteria are used to guide the decisions on the intervention content and how to implement the intervention within a particular setting [15, 17]. These criteria involve an assessment of: affordability; practicability; effectiveness and cost-effectiveness; acceptability; side-effects/safety; equity.	7-8/ 106-111
		5. Com B model: We have provided additional information on page 7.	The COM-B model [17] forms the centre of the Behaviour Change Wheel [15, 17] and assists with understanding the behaviour in context (Stage 1 of intervention development). The COM-B model hypothesises that capability (C), opportunity (O) and motivation (M) all interact and can explain behaviour (B) and can become the focus for the behaviour change intervention. Within the COM-B model	7/ 90-100

			capability refers to the person's psychological and physical capacity to engage in the target behaviour. Opportunity refers to the factors that are external to the individual and influence the potential success of the behaviour (i.e. the physical environment or the social environment). Motivation involves the psychological processes that can trigger and direct behaviour, including reflective and automatic motivation.	
		6. Behaviour change technique taxonomy version 1: We have added information to page 13.	In addition, the Behaviour Change Technique Taxonomy Version 1 [27] informed our choice of behaviour change techniques (step 7). The Behaviour Change Technique Taxonomy Version 1 [27] comprises 93 behaviour change techniques and can be used to identify intervention components, enabling the standardisation of terms as well as the comparison of behaviour change techniques across studies. Using the Behaviour Change Technique Taxonomy Version 1 [27] (which is a taxonomy of 93 behaviour change	13/ 225- 230 21/ 367- 373

			techniques) together with the findings from our systematic review, we selected the seven techniques we believed were most suitable to facilitate behaviour change and support prevention practices (information about social and environmental consequences; information on health consequences; feedback on behaviour; feedback on the outcome of the behaviour; prompts/cues; instruction on how to perform the behaviour; demonstration of behaviour).	
Reviewer: 2	This manuscript reports the development of a care bundle intended to help the prevention of pressure ulcers in UK Nursing Homes. The work is well described with the limitation of the small sample of participants clearly noted. There may be value in extending the background section to provide information upon the occurrence of pressure ulcers in UK nursing homes (where this data exists?).	We have provided an additional sentence within the background section demonstrating the occurrence of pressure ulcers in residential and nursing homes in a Northern UK city, based on a point prevalence survey (Hall et al., 2014).	Reducing and eliminating pressure injuries across all healthcare settings in the UK is a priority [5]. People at high risk of pressure injury include those who are seriously ill, the elderly and those with impaired mobility [6, 7]. Thus many people living in nursing homes are likely to be at an increased risk of pressure injury. Moreover, a point prevalence survey of complex wounds (e.g., pressure ulcers, leg ulcers) conducted in a northern UK city found 26% of individuals with a pressure ulcer (an	5/ 50-57

			open wound caused by pressure) lived in residential or nursing homes [8].	
Reviewer: 3	Thank you for submitting this manuscript about the development of a care bundle to prevent pressure ulcers in residents of a nursing home in North West England. I have several suggestions to strengthen this paper. Recently, the correct term is pressure injuries.	Thank you for raising this issue as it continues to be something that is debated within the literature. Originally we used the term pressure ulcer as that is the term most commonly used across Europe despite the NPUAP (2016) updated terminology. However, we have changed the term pressure ulcer to pressure injury in light of your comment.		
	Although the title refers to nursing home settings, it is clear that this care bundle development occurred in one nursing home. The study design was the development of a care bundle and this should be consistently presented and revised in the abstract.	We have revised this for consistency.	Design: The development of a care bundle.	2/ 5
	Additional limitations are exclusion of residents (please see recent Australian studies), residents' families, and multidisciplinary team members such as allied health professionals and physicians. It would be helpful to address the tension between "treating the whole patient rather than the hole in the patient" as care bundles may be interpreted as "treating the hole in the patient" by some.	We have added the exclusion of additional group (residents, families and multidisciplinary team members) as a limitation of the study and addressed the issue of treating the resident in a holistic manner within the future research section.	A limitation was the exclusion of residents and their families, as well as the wider multidisciplinary team (e.g., podiatrists, dieticians); and the inclusion of only one nursing home and the relatively small number of tissue viability nurse workshop participants. The next phase of this research is to test the feasibility of implementing the	29/ 450- 453 29- 30/ 465- 473

			care bundle in a nursing home context. If the care bundle intervention is feasible and acceptable to nursing home care staff, further evaluation will be necessary to assess the clinical and cost-effectiveness. The explicit theoretical links provided through the use of the Behaviour Change Wheel [15, 17] and Behaviour Change Technique Taxonomy Version 1 [27] will facilitate future replications and data synthesis. In addition, exploring the views of residents, their families and the wider multidisciplinary team will be vital to ensure that a holistic approach is taken to the prevention of pressure injuries in nursing home residents.	
	In the section about the Nominal Group process, further details about deciding on three bundle elements (rather than five) and then excluding the most popular elements would be helpful. These choices were perplexing given the percentages in the Table. Please explain why you decided against using an established skin assessment tool.	We have added information to the results section detailing the discussions held by participants and outlining our decision-making processes. In brief, we chose to exclude nutrition, hydration and continence care during these discussions for the following reasons:  - Nutrition and hydration interventions are not 	Whilst the participants deemed nutrition and hydration and continence care important, they agreed that only those residents with inadequate nutrition and hydration require additional nutrition and fluid [9]; therefore, this element would be redundant for some individuals (making the care bundle more of a checklist).	18/ 306- 315

		recommended for all people at risk of developing a pressure injury; only those with an inadequate nutrition and hydration status requiring additional nutrition and fluid. Consequently, this element would be irrelevant for many, and where it was relevant the primary motivation for correcting deficits would not be pressure injury prevention. The aim of a care bundle is to encourage effective behaviour change in clinical practice by grouping a small number of core behaviours that need to be delivered consistently and frequently, rather than to be an exhaustive checklist of all behaviours involved.  - Contenance care: During 	Participants believed that continence care was a separate, complex issue; requiring a number of detailed steps to prevent damage to skin integrity and likely to require its own care bundle [32]. Consequently participants decided that providing and monitoring such clinical interventions are part of basic care and should not be included in a specific pressure injury prevention bundle.	
--	--	--	---	--

		discussions it became clear that clinical partners (based on their clinical expertise and research evidence) felt that continence care should be viewed as a wider issue that needed its own care bundle (e.g., The Health Foundation's continence promotion care bundle, 2017). Participants were concerned that inclusion of continence care as a brief element within a pressure injury care bundle would underplay the complexity of continence care and reduce its importance. Skin assessment tool: Within the UK specific skin assessment tools are not usually used, rather health workers would conduct a risk assessment and there are specific tools for this. However, the tools used vary. We chose to include a formal risk assessment as an action conducted prior to delivering the care		
--	--	--	--	--

		bundle, as the outcome of this assessment will inform the minimum frequency with which a resident should receive the care bundle per day. Risk assessments are conducted in nursing homes on a monthly basis (except where a resident's health is changing rapidly) and so it was not deemed appropriate to include it as a specific element that needed to be conducted daily. The skin assessment was included within the skin 'inspection' element.		
	In Step 4, it is unclear who the 25 participants were and how these participants were recruited	We have added information to this section on page 12.	We purposively recruited individuals who provide care for those at risk of developing pressure injuries in nursing homes and collected data from 25 participants (healthcare assistants (n = 7), registered nurses (n = 11), nurse managers (n = 3) and community-based tissue viability nurses (n = 4)).	12/ 202- 205
	Further details about the analysis process with direct quotes would strengthen this section	We have added details to Step 4 in the methods section to explain that we conducted a deductive analysis of the qualitative data.	We analysed the data deductively, using the Theoretical Domains Framework and identified the behavioural and psychological influences on pressure injury prevention by mapping the salient barriers and facilitators identified onto the COM-B model, using the	12/ 213- 217 19/ 335- 338

		In the results section we are not able to duplicate material published elsewhere and have referred to the paper that details our findings.	guidance provided by the Behaviour Change Wheel [15]. The semi-structured interview data (reported elsewhere [25]), when mapped on to the COM-B model, suggested the following factors as influences on the prevention of pressure injury in nursing home settings: psychological and physical capability; physical and social opportunity; and reflective motivation.	
	Please attend to minor grammatical issues with correct use of commas, semi-colons and colons.	We have proof read the paper.		

VERSION 2 – REVIEW

REVIEWER	Zena Moore Royal College of Surgeon in Ireland
REVIEW RETURNED	31-Jan-2019

GENERAL COMMENTS	Many thanks for addressing the feedback, the responses have added clarity
---

REVIEWER	Donna E. Martin University of Manitoba
REVIEW RETURNED	07-Feb-2019

GENERAL COMMENTS	Thank you for sharing this important project and incorporating reviewers' suggestions into this revised manuscript. I look forward to seeing it published.
--